# CADet: Fully Self-Supervised Anomaly Detection With Contrastive Learning

**Charles Guille-Escuret**
ServiceNow Research, Mila,
Université de Montréal
guillech@mila.quebec

**Pau Rodriguez**
ServiceNow Research
pau.rodriguez@servicenow.com

**David Vazquez**
ServiceNow Research
david.vazquez@servicenow.com

**Ioannis Mitliagkas**
Mila, Université de Montréal,
Canada CIFAR AI chair
ioannis@mila.quebec

**Joao Monteiro**
ServiceNow Research
joao.monteiro@servicenow.com

## Abstract

Handling out-of-distribution (OOD) samples has become a major stake in the real-world deployment of machine learning systems. This work explores the use of self-supervised contrastive learning to the simultaneous detection of two types of OOD samples: unseen classes and adversarial perturbations. First, we pair self-supervised contrastive learning with the maximum mean discrepancy (MMD) two-sample test. This approach enables us to robustly test whether two independent sets of samples originate from the same distribution, and we demonstrate its effectiveness by discriminating between CIFAR-10 and CIFAR-10.1 with higher confidence than previous work. Motivated by this success, we introduce CADet (Contrastive Anomaly Detection), a novel method for OOD detection of single samples. CADet draws inspiration from MMD, but leverages the similarity between contrastive transformations of a same sample. CADet outperforms existing adversarial detection methods in identifying adversarially perturbed samples on ImageNet and achieves comparable performance to unseen label detection methods on two challenging benchmarks: ImageNet-O and iNaturalist. Significantly, CADet is fully self-supervised and requires neither labels for in-distribution samples nor access to OOD examples.[1]

## 1 Introduction

While modern machine learning systems have achieved countless successful real-world applications, handling out-of-distribution (OOD) inputs remains a tough challenge of significant importance. The

---

[1]Our code to compute CADet scores is publicly available as an OpenOOD fork at `https://github.com/charlesGE/OpenOOD-CADet`.

problem is especially acute for high-dimensional problems like image classification. Models are typically trained in a close-world setting but inevitably faced with novel input classes when deployed in the real world. The impact can range from displeasing customer experience to dire consequences in the case of safety-critical applications such as autonomous driving [31] or medical analysis [55]. Although achieving high accuracy against all meaningful distributional shifts is the most desirable solution, it is particularly challenging. An efficient method to mitigate the consequences of unexpected inputs is to perform anomaly detection, which allows the system to anticipate its inability to process unusual inputs and react adequately.

Anomaly detection methods generally rely on one of three types of statistics: features, logits, and softmax probabilities, with some systems leveraging a mix of these [66]. An anomaly score $f(x)$ is computed, and then detection with threshold $\tau$ is performed based on whether $f(x) > \tau$. The goal of a detection system is to find an anomaly score that efficiently discriminates between in-distribution and out-of-distribution samples. However, the common problem of these systems is that different distributional shifts will unpredictably affect these statistics. Accordingly, detection systems either achieve good performance on specific types of distributions or require tuning on OOD samples. In both cases, their practical use is severely limited. Motivated by these issues, recent work has tackled the challenge of designing detection systems for unseen classes without prior knowledge of the unseen label set or access to OOD samples [68, 63, 66].

We first investigate the use of maximum mean discrepancy two-sample test (MMD) [19] in conjunction with self-supervised contrastive learning to assess whether two sets of samples have been drawn from the same distribution. Motivated by the strong testing power of this method, we then introduce a statistic inspired by MMD and leveraging contrastive transformations. Based on this statistic, we propose CADet (Contrastive Anomaly Detection), which is able to detect OOD samples from single inputs and performs well on both label-based and adversarial detection benchmarks, *without requiring access to any OOD samples to train or tune the method.*

Only a few works have addressed these tasks simultaneously. These works either focus on particular in-distribution data such as medical imaging for specific diseases [65] or evaluate their performances on datasets with very distant classes such as CIFAR10 [32], SVHN [47], and LSUN [73], resulting in simple benchmarks that do not translate to general real world applications [33, 51].

**Contributions** Our main contributions are as follows:

- We use similarity functions learned by self-supervised contrastive learning with MMD to show that the test sets of CIFAR10 and CIFAR10.1 [52] have different distributions.
- We propose a novel improvement to MMD and show it can also be used to confidently detect distributional shifts when given a small number of samples.
- We introduce CADet, a fully self-supervised method for OOD detection inspired by MMD, and show it outperforms current methods in adversarial detection tasks while performing well on label-based OOD detection.

The outline is as follows: in Section 2, we discuss relevant previous work. Section 3 describes the self-supervised contrastive method based on SimCLRv2 [5] used in this work. Section 4 explores the application of learned similarity functions in conjunction with MMD to verify whether two independent sets of samples are drawn from the same distribution. Section 5 presents CADet and evaluates its empirical performance. Finally, we discuss results and limitations in Section 6.

## 2 Related work

We propose a self-supervised contrastive method for anomaly detection (both unknown classes and adversarial attacks) inspired by MMD. Thus, our work intersects with the MMD, label-based OOD detection, adversarial detection, and self-supervised contrastive learning literature.

**MMD** two-sample test has been extensively studied [19, 67, 18, 62, 8, 29], though it is, to the best of our knowledge, the first time a similarity function trained via contrastive learning is used in conjunction with MMD. Liu et al. [35] uses MMD with a deep kernel trained on a fraction of the samples to argue that CIFAR10 and CIFAR10.1 have different test distributions. We build upon that work by confirming their finding with higher confidence levels, using fewer samples. Dong et al. [11] explored applications of MMD to OOD detection.

**Label-based OOD detection methods** discriminate samples that differ from those in the training distribution. We focus on unsupervised OOD detection in this work, *i.e.*, we do not assume access to data labeled as OOD. Unsupervised OOD detection methods include density-based [74, 45, 46, 7, 12, 53, 59, 17, 37, 10], reconstruction-based [56, 75, 9, 50, 49, 7], one-class classifiers [57, 54], self-supervised [15, 25, 2, 63], and supervised approaches [34, 23], though some works do not fall into any of these categories [66, 60]. We refer to Yang et al. [71] for an overview of the many recent works in this field.

**Adversarial detection** discriminates adversarial samples from the original data. Adversarial samples are generated by minimally perturbing actual samples to produce a change in the model's output, such as a misclassification. Most works rely on the knowledge of some attacks for training [1, 43, 13, 39, 76, 48, 40], with the exception of [27].

**Self-supervised contrastive learning** methods [69, 22, 4, 5] are commonly used to pre-train a model from unlabeled data to solve a downstream task such as image classification. Contrastive learning relies on instance discrimination trained with a contrastive loss [21] such as infoNCE [20].

**Contrastive learning for OOD detection** aims to find good representations for detecting OOD samples in a supervised [36, 30] or unsupervised [68, 44, 58] setting. Perhaps the closest work in the literature is CSI [63], which found SimCLR features to have good discriminative power for unknown classes detection and leveraged similarities between transformed samples in their score. However, this method is not well-suited for adversarial detection. CSI ignores the similarities between different transformations of a same sample, an essential component to perform adversarial detection (see Section 6.2). In addition, CSI scales their score with the norm of input representations. While efficient on samples with unknown classes, it is unreliable on adversarial perturbations, which typically increase representation norms. Finally, we failed to scale CSI to ImageNet.

# 3   Contrastive model

We build our model on top of SimCLRv2 [5] for its simplicity and efficiency. It is composed of an encoder backbone network $f_\theta$ as well as a 3-layer contrastive head $h_{\theta'}$. Given an in-distribution sample $\mathcal{X}$, a similarity function *sim*, and a distribution of training transformations $\mathcal{T}_{train}$, the goal is to simultaneously maximize

$$\mathbb{E}_{x\sim\mathcal{X};t_0,t_1\sim\mathcal{T}_{train}}\left[sim(h_{\theta'}\circ f_\theta(t_0(x)), h_{\theta'}\circ f_\theta(t_1(x)))\right],$$

and minimize

$$\mathbb{E}_{x,y\sim\mathcal{X};t_0,t_1\sim\mathcal{T}_{train}}\left[sim(h_{\theta'}\circ f_\theta(t_0(x)), h_{\theta'}\circ f_\theta(t_1(y)))\right],$$

i.e., we want to learn representations in which random transformations of a same example are close while random transformations of different examples are distant.

To achieve this, given an input batch $\{x_i\}_{i=1,...,N}$, we compute the set $\{x_i^{(j)}\}_{j=0,1;i=1,...,N}$ by applying two transformations independently sampled from $\mathcal{T}_{train}$ to each $x_i$. We then compute the embeddings $z_i^{(j)} = h_{\theta'}\circ f_\theta(x_i^{(j)})$ and apply the following contrastive loss:

$$L(\mathbf{z}) = \sum_{i=1,...,N} -\log\frac{u_{i,j}}{\sum_{j\in\{1,...,N\}}(u_{i,j}+v_{i,j})}, \tag{1}$$

where

$$u_{i,j} = e^{sim(z_i^{(0)},z_j^{(1)})/\tau} \quad \text{and} \quad v_{i,j} = \mathbb{1}_{i\neq j}e^{sim(z_i^{(0)},z_j^{(0)})/\tau}.$$

$\tau$ is the temperature hyperparameter and $sim(x,y) = \frac{\langle x|y\rangle}{\|x\|_2\|y\|_2}$ is the *cosine* similarity.

**Hyperparameters:** We follow as closely as possible the setting from SimCLRv2 with a few modifications to adapt to hardware limitations. In particular, we use the LARS optimizer [72] with learning rate 1.2, momentum 0.9, and weight decay $10^{-4}$. Iteration-wise, we scale up the learning rate for the first 40 epochs linearly, then use an iteration-wise cosine decaying schedule until epoch 800, with temperature $\tau = 0.1$. We train on 8 $V100$ GPUs with an accumulated batch size of 1024. We compute the contrastive loss on all batch samples by aggregating the embeddings computed by each GPU. We use synchronized BatchNorm and fp32 precision and do not use a memory buffer. We use the same set of transformations, i.e., Gaussian blur and horizontal flip with probability 0.5, color

jittering with probability 0.8, random crop with scale uniformly sampled in $[0.08, 1]$, and grayscale with probability 0.2.

For computational simplicity and comparison with previous work, we use a ResNet50 encoder architecture with final features of size 2048. Following SimCLRv2, we use a three-layer fully connected contrastive head with hidden layers of width 2048 using ReLU activation and batchNorm and set the last layer projection to dimension 128. For evaluation, we use the features produced by the encoder without the contrastive head. We do not, at any point, use supervised fine-tuning.

## 4 MMD two-sample test

The **Maximum Mean Discrepancy (MMD)** is a statistic used in the MMD two-sample test to assess whether two sets of samples $S_\mathbb{P}$ and $S_\mathbb{Q}$ are drawn from the same distribution. It estimates the expected difference between the intra-set distances and the across-sets distances.

**Definition 4.1** (Gretton et al. [19])**.** Let $k : \mathcal{X} \times \mathcal{X} \to \mathbb{R}$ be the kernel of a reproducing Hilbert space $\mathcal{H}_k$, with feature maps $k(\cdot, x) \in \mathcal{H}_k$. Let $X, X' \sim \mathbb{P}$ and $Y, Y' \sim \mathbb{Q}$. Under mild integrability conditions,

$$MMD(\mathbb{P}, \mathbb{Q}; \mathcal{H}_k) := \sup_{f \in \mathcal{H}, \|f\|_{\mathcal{H}_k} \leq 1} |\mathbb{E}[f(X)] - \mathbb{E}[f(Y)]| \tag{2}$$

$$= \sqrt{\mathbb{E}[k(X, X') + k(Y, Y') - 2k(X, Y)]}. \tag{3}$$

Given two sets of $n$ samples $S_\mathbb{P} = \{X_i\}_{i \leq n}$ and $S_\mathbb{Q} = \{Y_i\}_{i \leq n}$, respectively drawn from $\mathbb{P}$ and $\mathbb{Q}$, we can compute the following unbiased estimator [35]:

$$\widehat{MMD}_u^2(S_\mathbb{P}, S_\mathbb{Q}; k) := \frac{1}{n(n-1)} \sum_{i \neq j} (k(X_i, X_j) + k(Y_i, Y_j) - k(X_i, Y_j) - k(Y_i, X_j)). \tag{4}$$

Under the null hypothesis $\mathfrak{h}_0 : \mathbb{P} = \mathbb{Q}$, this estimator follows a normal distribution of mean 0 [19]. Its variance can be directly estimated [18], but it is simpler to perform a permutation test as suggested in Sutherland et al. [62], which directly yields a $p$-value for $\mathfrak{h}_0$. The idea is to use random splits $X, Y$ of the input sample sets to obtain $n_{perm}$ different (though not independent) samplings of $\widehat{MMD}_u^2(X, Y; k)$, which approximate the distribution of $\widehat{MMD}_u^2(S_\mathbb{P}, S_\mathbb{P}; k)$ under the null hypothesis.

Liu et al. [35] trains a deep kernel to maximize the test power of the MMD two-sample test on a training split of the sets of samples to test. We propose instead to use our learned similarity function without any fine-tuning. Note that we return the $p$-value $\frac{1}{n_{perm}+1}\left(1 + \sum_{i=1}^{n_{perm}} \mathbb{1}(p_i \geq est)\right)$ instead of $\frac{1}{n_{perm}} \sum_{i=1}^{n_{perm}} \mathbb{1}(p_i \geq est)$. Indeed, under the null hypothesis $\mathbb{P} = \mathbb{Q}$, $est$ and $p_i$ are drawn from the same distribution, so for $j \in \{0, 1, \ldots, n_{perm}\}$, the probability for $est$ to be smaller than exactly $j$ elements of $\{p_i\}$ is $\frac{1}{n_{perm}+1}$. Therefore, the probability that $j$ elements or less of $\{p_i\}_i$ are larger than $est$ is $\sum_{i=0}^{j} \frac{1}{n_{perm}+1} = \frac{j+1}{n_{perm}+1}$. While this change has a small impact for large values of $n_{perm}$, it is essential to guarantee that we indeed return a correct $p$-value. Notably, the algorithm of Liu et al. [35] has a probability $\frac{1}{n_{perm}} > 0$ to return an output of 0.00 even under the null hypothesis.

Additionnally, we propose a novel version of MMD called MMD-CC (MMD with Clean Calibration). Instead of computing $p_i$ based on random splits of $S_\mathbb{P} \bigcup S_\mathbb{Q}$, we require as input two disjoint sets of samples drawn from $\mathbb{P}$ and compute $p_i$ based on random splits of $S_\mathbb{P}^{(1)} \bigcup S_\mathbb{P}^{(2)}$ (see Algorithm 1). This change requires to use twice as many samples from $\mathbb{P}$, but reduces the variance induced by the random splits of $S_\mathbb{P} \bigcup S_\mathbb{Q}$, which is significant when the number of samples is small. Note that $S_\mathbb{P}^{(1)}$, $S_\mathbb{P}^{(2)}$ and $S_\mathbb{Q}$ must always have the same size. Under the null hypothesis, $\mathbb{P} = \mathbb{Q}$, $S_\mathbb{P}^{(1)}$, $S_\mathbb{P}^{(2)}$ and $S_\mathbb{Q}$ are identically distributed, so the $p_i$ conserve the same distribution as for MMD (this is not the case when $\mathbb{P} \neq \mathbb{Q}$, hence the difference in testing power). Thus, the validity of MMD-CC follows from the validity of MMD.

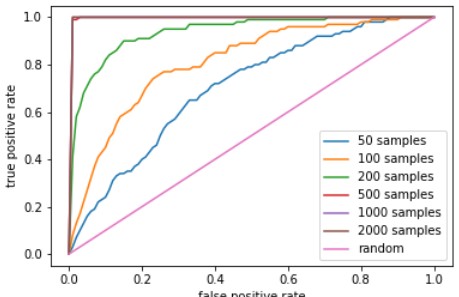 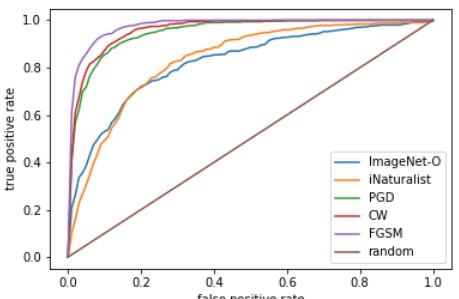

(a) ROC curves of MMD two-sample test on CIFAR-10.1 against CIFAR-10 for different sample sizes.

(b) ROC curves of CADet on different distribution shifts of ImageNet.

---

**Algorithm 1** MMD-CC two-sample test

---

**Input:** $S_{\mathbb{P}}^{(1)}, S_{\mathbb{P}}^{(2)}, S_{\mathbb{Q}}, n_{perm}, sim$

$est \leftarrow \widehat{MMD}_u^2(S_{\mathbb{P}}^{(1)}, S_{\mathbb{Q}}; sim)$
**for** $i = 1, 2, \ldots, n_{perm}$ **do**
   Randomly split $S_{\mathbb{P}}^{(1)} \bigcup S_{\mathbb{P}}^{(2)}$ into two disjoint sets $X, Y$ of equal size
   $p_i \leftarrow \widehat{MMD}_u^2(X, Y; sim)$
**end for**
**Output:** $p$: $\frac{1}{1+n_{perm}} \left(1 + \sum_{i=1}^{n_{perm}} \mathbb{1}\left(p_i \geq est\right)\right)$

---

## 4.1 Distribution shift between CIFAR-10 and CIFAR-10.1 test sets

After years of evaluation of popular supervised architectures on the test set of CIFAR-10 [32], modern models may overfit it through their hyperparameter tuning and structural choices. CIFAR-10.1 [52] was collected to verify the performances of these models on a *truly* independent sample from the training distribution. The authors note a consistent drop in accuracy across models and suggest it could be due to a distributional shift, though they could not demonstrate it. Recent work [35] leveraged the two-sample test to provide strong evidence of distributional shifts between the test sets of CIFAR-10 and CIFAR-10.1. We run MMD-CC and MMD two-sample tests for 100 different samplings of $S_{\mathbb{P}}^{(1)}, S_{\mathbb{P}}^{(2)}, S_{\mathbb{Q}}$, using every time $n_{perm} = 500$, and rejecting $\mathfrak{h}_0$ when the obtained $p$-value is below the threshold $\alpha = 0.05$. We also report results using cosine similarity applied to the features of supervised models as a comparative baseline. We report the results in Table 1 for a range of sample sizes. We compare the results to three competitive methods reported in Liu et al. [35]: Mean embedding (ME) [8, 29], MMD-D [35], and C2ST-L [6]. Finally, we show in Figure 1a the ROC curves of the proposed model for different sample sizes.

Other methods in the literature do not use external data for pre-training, as we do with ImageNet, which makes a fair comparison difficult. However, it is noteworthy that our learned similarity can very confidently distinguish samples from the two datasets, even in settings with fewer samples available. Furthermore, while we achieve excellent results even with a supervised network, our model trained with contrastive learning outperforms the supervised alternative very significantly. We note however that with such high number of samples available, MMD-CC performs slightly worse than MMD. Finally, we believe the confidence obtained with our method decisively concludes that CIFAR10 and CIFAR10.1 have different distributions, which is likely the primary explanation for the significant drop in performances across models on CIFAR10.1, as conjectured by Recht et al. [52]. The difference in distribution between CIFAR10 and CIFAR10.1 is neither based on label set nor adversarial perturbations, making it an interesting task.

Table 1: Average rejection rates of $\hbar_0$ on CIFAR-10 vs CIFAR-10.1 for $\alpha = 0.05$ across different sample sizes $n$, using a ResNet50 backbone.

|  | n=2000 | n=1000 | n=500 | n=200 | n=100 | n=50 |
|---|---|---|---|---|---|---|
| ME [8] | 0.588 | - | - | - | - | - |
| C2ST-L [6] | 0.529 | - | - | - | - | - |
| MMD-D [35] | 0.744 | - | - | - | - | - |
| MMD + SimCLRv2 (ours) | **1.00** | **1.00** | 0.997 | 0.702 | 0.325 | 0.154 |
| MMD-CC + SimCLRv2 (ours) | **1.00** | **1.00** | 0.997 | 0.686 | 0.304 | 0.150 |
| MMD + Supervised (ours) | **1.00** | **1.00** | 0.884 | 0.305 | 0.135 | 0.103 |
| MMD-CC + Supervised (ours) | **1.00** | **1.00** | 0.870 | 0.298 | 0.131 | 0.096 |

Table 2: AUROC for detection using two-sample test on 3 to 20 samples drawn from ImageNet and from ImageNet-O, iNaturalist or PGD perturbations, with a ResNet50 backbone.

|  | ImageNet-O | | | | iNaturalist | | | | PGD | | | |
|---|---|---|---|---|---|---|---|---|---|---|---|---|
| n_samples | 3 | 5 | 10 | 20 | 3 | 5 | 10 | 20 | 3 | 5 | 10 | 20 |
| MMD + SimCLRv2 | 64.3 | 72.4 | 86.9 | 97.6 | 88.3 | 97.6 | **99.5** | **99.5** | 35.2 | 53.8 | 86.6 | 98.8 |
| MMD-CC + SimCLRv2 | **65.3** | **73.2** | **88.0** | **97.7** | 95.4 | 99.2 | **99.5** | **99.5** | **70.5** | **84.0** | **96.6** | **99.5** |
| MMD + Supervised | 62.7 | 69.7 | 83.2 | 96.4 | 91.8 | 98.7 | **99.5** | **99.5** | 20.0 | 22.5 | 33.0 | 57.5 |
| MMD-CC + Supervised | 62.6 | 71.0 | 85.5 | 97.2 | **98.0** | **99.5** | **99.5** | **99.5** | 57.4 | 61.3 | 70.5 | 85.8 |

## 4.2 Detection of distributional shifts from small number of samples

Given a small set of samples with potential unknown classes or adversarial attacks, we can similarly use the two-sample test with our similarity function to verify whether these samples are in-distribution [14]. In particular, we test for samples drawn from ImageNet-O, iNaturalist, and PGD perturbations, with sample sizes ranging from 3 to 20. For these experiments, we sample $S_{\mathbb{P}}^{(1)}$ and $S_{\mathbb{P}}^{(2)}$ 5000 times across all of ImageNet's validation set and compare their MMD and MMD-CC estimators to the one obtained from $S_{\mathbb{P}}$ and $S_{\mathbb{Q}}$. We report in Table 2 the AUROC of the resulting detection and compare it to the ones obtained with a supervised ResNet50 as the baseline.

Such a setting where we use several samples assumed to be drawn from a same distribution to perform detection is uncommon, and we are not aware of prior baselines in the literature. Despite using very few samples ($3 \leq n \leq 20$), our method can detect OOD samples with high confidence. We observe particularly outstanding performances on iNaturalist, which is easily explained by the fact that the subset we are using (cf. Section 1) only contains plant species, logically inducing an abnormally high similarity within its samples. Furthermore, we observe that MMD-CC performs significantly better than MMD, especially on detecting samples perturbed by PGD.

Although our method attains excellent detection rates for sufficient numbers of samples, the requirement to have a set of samples all drawn from the same distribution to perform the test makes it unpractical for real-world applications. In the following section, we present CADet, a detection method inspired by MMD but applicable to anomaly detection with single inputs.

## 5 CADet: Contrastive Anomaly Detection

While the numbers in Section 4 demonstrate the reliability of two-sample test coupled with contrastive learning for identifying distributional shifts, it requires several samples from the same distribution, which is generally unrealistic for practical detection purposes. This section presents CADet, a method to leverage contrastive learning for OOD detection on single samples.

Self-supervised contrastive learning trains a similarity function $s$ to maximize the similarity between augmentations of the same sample, and minimize the similarity between augmentations of different samples. Given an input sample $x^{test}$, we propose to leverage this property to perform anomaly detection on $x^{test}$, taking inspiration from MMD two-sample test. More precisely, given a transformation distribution $\mathcal{T}_{val}$, we compute $n_{trs}$ random transformations $x_i^{test}$ of $x_{test}$, as well as $n_{trs}$ random transformations $x_i^{(k)}$ on each sample $x^{(k)}$ of a held-out validation dataset $X_{val}^{(1)}$. We then compute the

intra-similarity and out-similarity:

$$m^{in}(x^{test}) := \frac{\sum\limits_{i \neq j} s(x_i^{test}, x_j^{test})}{n_{trs}(n_{trs}+1)}, \quad \text{and} \quad m^{out}(x^{test}) := \frac{\sum\limits_{x^{(k)} \in X_{val}^{(1)}} \sum\limits_{i,j} s(x_i^{test}, x_j^{(k)})}{n_{trs}^2 \times |X_{val}^{(1)}|}. \quad (5)$$

We finally define the following statistic to perform detection:

$$score_C := m^{in} + \gamma m^{out}. \quad (6)$$

**Intuition:**   note that while CADet is inspired by MMD, it has some significant differences. $m_{in}$ is computed across transformations of a same sample, while $m_{out}$ is computed across transformations and across samples. Therefore, even under the null hypothesis, $m_{in}$ and $m_{out}$ cannot be expected to have the same distribution. Besides, both score can be expected to be smaller on OOD samples. Indeed, OOD samples can be expected to have representations further from clean data in average, thus decreasing $m_{out}$. Furthermore, since the self-supervised model was trained to maximize the similarity between transformations of a same sample, it is reasonable to expect that the model will perform this task better in-distribution than out-of-distribution, and thus that $m_{in}$ will be higher in-distribution than on OOD data. This intuition justifies adding $m_{in}$ to $m_{out}$, instead of the subtraction operated by MMD. Finally, note that we do not consider the intra-similarity between validation samples: the validation is fixed, so it is necessarily a constant.

**Calibration:**   since we do not assume knowledge of OOD samples, it is difficult *a priori* to tune $\gamma$, although crucial to balance information between intra-sample similarity and cross-sample similarity. As a workaround, we calibrate $\gamma$ by equalizing the variance between $m^{in}$ and $\gamma m^{out}$ on a second set of validation samples $X_{val}^{(2)}$:

$$\gamma = \sqrt{\frac{Var\{m^{in}(x), x \in X_{val}^{(2)}\}}{Var\{m^{out}(x), x \in X_{val}^{(2)}\}}}. \quad (7)$$

It is important to note that the calibration of $\gamma$ is essential, because $m_{in}$ and $m_{out}$ can have very different means and variances. Since $m_{out}$ measures the out-similarity between transformations of *different* samples, it can be expected to be low in average, with high variance. On the contrary, $m_{in}$ measures the similarity between transformations of a *same* sample, and can thus be expected to have higher mean and lower variance, which is confirmed in Table 5. Therefore, simply setting $\gamma = 1$ would likely results in one of the score dominating the other.

Rather than evaluating the false positive rate (FPR) for a range of possible thresholds $\tau$, we use the hypothesis testing approach to compute the p-value:

$$p_v(x^{test}) = \frac{|\{x \in X_{val}^{(2)} \; s.t. \; score_C(x) < score_C(x^{test})\}| + 1}{|\{X_{val}^{(2)}\}| + 1}. \quad (8)$$

 The full pseudo-codes of the calibration and testing steps are given ins Appendix A. Setting a threshold $p \in [0,1]$ for $p_v$ will result in a FPR of mean $p$, with a variance dependant of $|X_{val}^{(2)}|$.

Section 5.1 further describes our experimental setting.

## 5.1   Experiments

For all evaluations, we use the same transformations as SimCLRv2 except color jittering, Gaussian blur and grayscaling. We fix the random crop scale to 0.75. We use $|\{X_{val}^{(2)}\}| = 2000$ in-distribution samples, $|\{X_{val}^{(1)}\}| = 300$ separate samples to compute cross-similarities, and 50 transformations per sample. We pre-train a ResNet50 with ImageNet as in-distribution.

**Unknown classes detection:**  we use two challenging benchmarks for the detection of unknown classes. iNaturalist using the subset in [28] made of plants with classes that do not intersect ImageNet. Wang et al. [66] noted that this dataset is particularly challenging due to proximity of its classes. We also evaluate on ImageNet-O [26]; explicitly designed to be challenging for OOD detection with ImageNet as in-distribution. We compare to recent works and report the AUROC scores in Table 3.

Table 3: AUROC for OOD detection on ImageNet-O and iNaturalist with ResNet50 backbone.

| | Training | iNaturalist | ImageNet-O | Average |
|---|---|---|---|---|
| MSP [23] | | 88.58 | 56.13 | 72.36 |
| Energy [38] | | 80.50 | 53.95 | 67.23 |
| ODIN [34] | | 86.48 | 52.87 | 69.68 |
| MaxLogit [24] | | 86.42 | 54.39 | 70.41 |
| KL Matching [24] | Supervised | 90.48 | 67.00 | 78.74 |
| ReAct [61] | | 87.27 | 68.02 | 77.65 |
| Mahalanobis [33] | | 89.48 | 80.15 | 84.82 |
| Residual [66] | | 84.63 | 81.15 | 82.89 |
| ViM [66] | | 89.26 | 81.02 | **85.14** |
| **CADet (ours)** | Supervised | **95.28** | 70.73 | 83.01 |
| | Self-supervised (contrastive) | 83.42 | **82.29** | 82.86 |

Table 4: AUROC for adversarial detection on ImageNet against PGD, CW and FGSM attacks, with ResNet50 backbone.

| | Tuned on Adv | Training | PGD | CW | FGSM | Average |
|---|---|---|---|---|---|---|
| ODIN [34] | Yes | Supervised | 62.30 | 60.29 | 68.10 | 63.56 |
| | | Contrastive | 59.91 | 60.23 | 64.99 | 61.71 |
| FS [70] | Yes | Supervised | 91.39 | 87.77 | 95.12 | 91.43 |
| | | Contrastive | 94.71 | 88.00 | 95.98 | 92.90 |
| LID [41] | Yes | Supervised | 93.13 | 91.08 | 93.55 | 92.59 |
| | | Contrastive | 91.94 | 89.50 | 93.56 | 91.67 |
| Hu [27] | Yes | Supervised | 84.31 | 84.29 | 77.95 | 82.18 |
| | | Contrastive | 94.80 | 95.19 | 78.18 | 89.39 |
| Hu [27] + self-calibration | **No** | Supervised | 66.40 | 59.58 | 71.02 | 65.67 |
| | | Contrastive | 75.69 | 75.74 | 69.20 | 73.54 |
| Mahalanobis [33] | **No** | Supervised | 92.71 | 91.01 | 96.92 | 93.55 |
| | | Contrastive | 84.14 | 83.78 | 79.90 | 82.61 |
| **CADet (ours)** | **No** | Supervised | 75.25 | 71.02 | 83.45 | 76.57 |
| | | Contrastive | **94.88** | **95.93** | **97.56** | **96.12** |

For Mahalanobis [33], since tuning on ood samples is not permitted, we use as score the mahalanobis score of the last layer as done in ViM [66]. Furthermore, to isolate the effect of the training scheme from the detection method, we present in Appendix B the performance of previous methods using the same self-supervised backbone as CADet, in conjunction with a trained linear layer.

**Adversarial detection:** for adversarial detection, we generate adversarial attacks on the validation partition of ImageNet against a pre-trained ResNet50 using three popular attacks: PGD [42], CW [3], and FGSM [16]. We follow the tuning suggested by Abusnaina et al. [1], i.e. PGD: norm $L_\infty$, $\delta = 0.02$, step size 0.002, 50 iterations; CW: norm $L_2$, $\delta = 0.10$, learning rate of 0.03, and 50 iterations; FGSM: norm $L_\infty$, $\delta = 0.05$. We compare our results with ODIN [34], which achieves good performances in Lee et al. [33] despite not being designed for adversarial detection, Feature Squeezing (FS) [70], Local Intrinsinc Dimensionality (LID) [41], Hu et al. [27], and Mahalanobis [33] (using the mahalanobis score of the last layer as for unknwon classes detection). Most existing adversarial detection methods assume access to adversarial samples during training (see Section 2). We additionally propose a modification to Hu et al. [27] to perform auto-calibration based on the mean and variance of the criterions on clean data, similarly to CADet's calibration step. We report the AUROC scores in Table 4 and illustrate them with ROC curves against each anomaly type in Figure 1b.

Table 5: Mean and variance of $m^{in}$ and $m^{out}$.

| | | IN-1K | iNat | IN-O | PGD | CW | FGSM |
|---|---|---|---|---|---|---|---|
| Mean | $m^{in}$ | 0.972 | 0.967 | 0.969 | 0.954 | 0.954 | 0.948 |
| | $m^{out}$ | 0.321 | 0.296 | 0.275 | 0.306 | 0.302 | 0.311 |
| | $\gamma m^{out}$ | 0.071 | 0.066 | 0.061 | 0.068 | 0.067 | 0.069 |
| Var | $m^{in}$ | 8.3e-05 | 7.8e-05 | 1.0e-04 | 2.1e-04 | 2.0e-04 | 2.1e-04 |
| | $m^{out}$ | 1.7e-03 | 7.0e-04 | 2.3e-03 | 7.0e-04 | 1.7e-03 | 1.1e-03 |

## 6 Discussion

### 6.1 Results

CADet performs particularly well on adversarial detection, surpassing alternatives by a wide margin. We argue that self-supervised contrastive learning is a suitable mechanism for detecting classification attacks due to its inherent label-agnostic nature. Interestingly, Hu et al. [27] also benefits from contrastive pre-training, achieving much higher performances than with a supervised backbone. However, it is very reliant on calibrating on adversarial samples, since we observe a significant drop in performances with auto-calibration.

We explain the impressive performances of CADet on adversarial detection by the fact that adversarial perturbations will negatively affect the capability of the self-supervised model to match different transformations of the image. In comparison, the model has high capability on clean data, having been trained specifically with that objective.

While CADet does not outperform existing methods on label-based OOD detection, it performs comparably, an impressive feat of generality considering adversarially perturbed samples and novel labels have significantly different properties, and that CADet does not require any tuning on OOD samples.

Notably, applying CADet to a supervised network achieves state-of-the-art performances on iNaturalist with ResNet50 architecture, suggesting CADet can be a reasonable standalone detection method on some benchmarks, independently of contrastive learning. In addition, the poor performances of the supervised network on ImageNet-O and adversarial attacks show that contrastive learning is essential to address the trade-off between different type of anomalies.

Overall, our results show CADet achieves an excellent trade-off when considering both adversarial and label-based OOD samples.

### 6.2 The predictive power of in-similarities and out-similarities

Table 5 reports the mean and variance of $m^{in}$ and $m^{out}$, and the rescaled mean $\gamma m^{out}$ across all distributions. Interestingly, we see that out-similarities $m^{out}$ better discriminate label-based OOD samples, while in-similarities $m^{in}$ better discriminate adversarial perturbations. Combining in-similarities and out-similarities is thus an essential component to simultaneously detect adversarial perturbations and unknown classes.

### 6.3 Limitations

**Computational cost:** To perform detection with CADet, we need to compute the features for a certain number of transformations of the test sample, incurring significant overhead. Figure 2 shows that reducing the number of transformations to minimize computational cost may not significantly affect performances. While the cal-

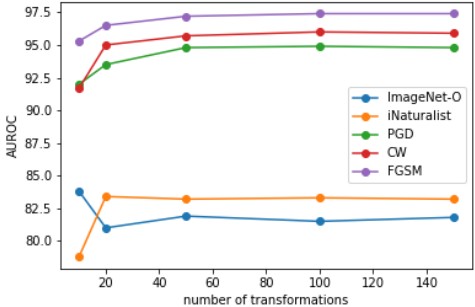

Figure 2: AUROC score of CADet against the number of transformations.

ibration step can seem expensive, it only required less than 10 minutes on a single A100 GPU, which is far from prohibitive. Moreover, it only needs to be run once for a given in-distribution. The

coefficient $\gamma$ and scores are all one-dimensional values that can be easily stored, and we purposely use a small number of validation samples $|\{X_{val}^{(1)}\}| = 300$ to make their embedding easy to memorize.

**Architecture scale:** as self-supervised contrastive learning is computationally expensive, we only evaluated our method on a ResNet50 architecture. In Wang et al. [66], the authors achieve significantly superior performances when using larger, recent architectures. The performances achieved with a ResNet50 are insufficient for real-world usage, and the question of how our method would scale to larger architectures remains open.

**Adaptive attacks:** we do not evaluate on adaptive attacks, and as such our detection method should be considered vulnerable to them. However, as shown in Tramer et al. [64], modern detection methods are typically weak to adaptive attacks. Since robustness to *any* adaptive attack is impossible to prove, we contend this is a limitation of detection methods in general, unspecific to CADet.

### 6.4 Future directions

While spurious correlations with background features are a problem in supervised learning, it is aggravated in self-supervised contrastive learning, where background features are highly relevant to the training task. We conjecture the poor performances of CADet on iNaturalist OOD detection are explained by the background similarities with ImageNet images, obfuscating the differences in relevant features. A natural way to alleviate this issue is to incorporate background transformations to the training pipeline, as was successfully applied in Ma et al. [40]. This process would come at the cost of being unable to detect shifts in background distributions, but such a case is generally less relevant to deployed systems. We leave to future work the exploration of how background transformations could affect the capabilities of CADet.

### 6.5 Conclusion

We have presented CADet, a method for both OOD and adversarial detection based on self-supervised contrastive learning. CADet achieves an excellent trade-off in detection power across different anomaly types, without requiring tuning on OOD samples. Additionally, we discussed how MMD could be leveraged with contrastive learning to assess distributional discrepancies between two sets of samples, and proved with high confidence that CIFAR10 and CIFAR10.1 have different distributions.

## Acknowledgments and Disclosure of Funding

We thank Issam Laradji for his precious help in fixing cluster issues and our NeurIPS 2023 reviewers and AC for their constructive criticism and for taking the time to engage in discussions. We believe their feedback has significantly improved the quality of our submission.

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

# A   Algorithms pseudo-code

We give the full pseudo codes for CADet's calibration step and inference step in Algorithm 2 and 3.

---

**Algorithm 2** CADet calibration step

---

**Input:** $X_{val}^{(1)}$, $X_{val}^{(2)}$, $\mathcal{T}_{eval}$, learned similarity function $s$, various hyperparameters used below;

1: **for** $x^{(1)} \in X_{val}^{(1)}$ **do**
2:    **for** $i = 1, 2, \ldots, n_{trs}$ **do**
3:       Sample $t$ from $\mathcal{T}_{eval}$
4:       $x_i^{(1)} \leftarrow t(x^{(1)})$
5:    **end for**
6: **end for**
7: $k \leftarrow 0$
8: **for** $x^{(2)} \in X_{val}^{(2)}$ **do**
9:    **for** $i = 1, 2, \ldots, n_{trs}$ **do**
10:       Sample $t$ from $\mathcal{T}_{eval}$
11:       $x_i^{(2)} \leftarrow t(x^{(2)})$
12:    **end for**
13:    $m_k^{in} \leftarrow \sum\limits_{i \neq j}^{n_{trs}} s(x_i^{(2)}, x_j^{(2)})$
14:    $m_k^{out} \leftarrow \sum\limits_{x^{(1)} \in X_{val}^{(1)}} \sum\limits_{i,j}^{n_{trs}} s(x_i^{(1)}, x_j^{(2)})$
15:    $k \leftarrow k + 1$
16: **end for**
17: $m^{in} \leftarrow \frac{m^{in}}{n_{trs}(n_{trs}+1)}$
18: $m^{out} \leftarrow \frac{m^{out}}{n_{trs}^2 \#\{X_{val}^{(1)}\}}$
19: $V_{in}, V_{out} = Var(m^{in}), Var(m^{out})$
20: $\gamma \leftarrow \sqrt{\frac{V_{in}}{V_{out}}}$
21: $k \leftarrow 0$
22: **for** $x^{(2)} \in X_{val}^{(2)}$ **do**
23:    $score_k \leftarrow m_k^{in} + \gamma \times m_k^{out}$
24:    $k \leftarrow k + 1$
25: **end for**

      **Output:** coefficient: $\gamma$, scores: $score_k$, transformed samples: $x_i^{(1)}$

---

**Algorithm 3** CADet testing step

---

**Input:** transformed samples: $x_i^{(1)}$, scores: $scores$, test sample: $x^{test}$, coefficient: $\gamma$, trasnformation set: $\mathcal{T}_{eval}$;

1: **for** $i = 1, 2, \ldots, n_{trs}$ **do**
2:     Sample $t$ from $\mathcal{T}_{eval}$
3:     $x_i^{test} \leftarrow t(x^{test})$
4: **end for**
5: $m^{in} \leftarrow \sum_{i \neq j}^{n_{trs}} s(x_i^{test}, x_j^{test})$
6: $m^{out} \leftarrow \sum_{x^{(1)} \in X_{val}^{(1)}} \sum_{i,j}^{n_{trs}} s(x_i^{test}, x_j^{(1)})$
7: $score^{te} \leftarrow \frac{m^{in}}{n_{trs}(n_{trs}+1)} + \gamma \frac{m^{out}}{n_{trs}^2 \#\{X_{val}^{(1)}\}}$
8: $rank \leftarrow 0$
9: **for** $score^{val} \in scores$ **do**
10:     **if** $score^{val} < score^{te}$ **then**
11:         $rank \leftarrow rank + 1$
12:     **end if**
13: **end for**
14: $p_{val} \leftarrow \frac{rank+1}{\#\{scores\}+1}$

**Output:** p-value: $p_{val}$

---

## B  Ablation on backbone

We present in table 6 the AUROC of previous methods on iNaturalist and ImageNet-O using a self-supervised contrastive ResNet50 backbone, as an ablation to table 3. Since most methods require class logits, we train a linear layer on top of the self-supervised features using all training labels. Thus, these methods still possess an unfair advantage over CADet by exploiting label information.

Table 6: AUROC for OOD detection on ImageNet-O and iNaturalist with ResNet50 backbone using self-supervised contrastive learning.

| | Training | iNaturalist | ImageNet-O | Average |
|---|---|---|---|---|
| MSP [23] | | 76.93 | 54.09 | 65.51 |
| Energy [38] | | 69.55 | 57.91 | 63.73 |
| ODIN [34] | | 81.83 | 52.10 | 66.97 |
| MaxLogit [24] | | 75.62 | 52.35 | 63.99 |
| KL Matching [24] | Self-supervised (contrastive) | 79.27 | 67.71 | 73.49 |
| ReAct [61] | | 78.95 | 68.08 | 73.52 |
| Mahalanobis [33] | | 72.60 | 76.49 | 74.55 |
| Residual [66] | | 78.60 | 80.99 | 79.80 |
| ViM [66] | | 72.17 | 85.11 | 78.64 |

## C  Tuning of baselines

For previous works, when not specified otherwise we use the same tuning strategies described in the original papers.

For Mahalanobis [33] we employ the score of the last layer only as done in ViM [66]. For LID [41], we try for $k$ in $[10, 20, ..., 100]$ (and find $k = 20$ to perform best on all benchmarks) and $\sigma$ in the range $[0, 10]$. For ODIN, for OOD detection, we use the default values described in Liang et al. [34] (since we assume no access to ood samples), and tune them for adversarial detection. For FS [70], we use the best joint detection of multiple squeezers, which was tuned on ImageNet, ie. Bit Depth 5-bit, Median Smoothing 2x2, and Non-Local Mean 11-3-4.

