# OpenReview forum: "CADet: Fully Self-Supervised Out-Of-Distribution Detection With Contrastive Learning"
_NeurIPS.cc/2023/Conference — NeurIPS 2023 poster_

### Official Review · Reviewer_hQ4Y · 2023-06-12

**Soundness:** 3 good
**Presentation:** 3 good
**Contribution:** 3 good
**Rating:** 5
**Confidence:** 5

**Summary:**

This paper consists of two mostly unrelated parts.  In the first part, the authors aim to detect whether two sets of samples are drawn from the same distribution.  To do this, they apply the unbiased estimate of the MMD, in the manner suggested by Sutherland et al., to produce a conformal p-value.  Then they propose a small variation on Sutherland's approach, which they call MMD-CC, that requires twice as many samples but supposedly reduces some variance (although no further explanation or analysis is given).  They apply these techniques to detecting distribution shift between CIFAR-10 and CIFAR-10.1 using a feature-generating network trained on an external dataset (ImageNet), similar to "outlier exposure" methods.

In the second part of the paper, they aim to detect whether a single sample is drawn from same distribution as a set of inliers or not.  For this, they propose the CADet method, which has no discernable relationship to MMD (3) or the unbiased MMD estimate in (4).  Rather, CADet is closer to the CSI approach from Tack et al. and the ICLR'21 approach of Sohn et al., in that it relies on contrastive learning and distributional shifts.  Because key parts of CADet are never explained by the authors, it's difficult to summarize further.  The authors test CADet on experiments where ImageNet is the inliers and other datsets are the outliers, but they never compare to the state-of-the-art CSI approach even though they write "the closest work in the literature is CSI" on line 86.

**Strengths:**

The combination of Sutherland's MMD with SimCLRv2 gives good empirical performance in detecting distributional shift between CIFAR-10 and CIFAR-10.1.

**Weaknesses:**

1. Claims about the existing literature are overblown and key citations are missing.
a. On line 46, the authors claim that "only a few works detect OOD samples from single inputs ... without requiring access to OOD samples to train or tune the method."  But there are dozens, if not hundreds, of papers proposing such methods.
b. Although the authors acknowledge that the CSI paper by Tack et al. is very close in spirit to CADet, they do not cite an even closer method in the ICLR'21 paper by Sohn et al.
c. Although the authors frequently compute conformal p-values, they don't cite any relevant literature.
d. The authors do not cite previous work that uses MMD for OOD, such as Dong et al.'s "Neural Mean Discrepancy for Efficient Out-of-Distribution Detection".

2. Throughout the paper, the p-values are invalid.  In the best case, the conformal p-values computed by the authors hold only only marginally, i.e., in expectation over the validation data.  They do NOT hold for a fixed validation set.  For a detailed explanation, see Bates et al. "Testing for Outliers with Conformal p-values" or Magesh et al. "Multiple testing framework for out-of-distribution detection" or Angelopoulos et al. "Conformal Prediction: A Gentle Introduction".

3. In some cases, the p-values are invalid for more fundamental reasons.  In MMD-CC Algorithm 1, for example, the conformal p-values are computed as if n_perm was the size of an i.i.d. validation dataset.  But n_perm permutations are not independent and so the p-values computed in this way are invalid.  For example, there is nothing preventing the user from specifing a value of n_perm that is higher than the total number of possible permutations.  In fact, the MMD-CC method is entirely heuristic and comes with no analysis, which is a serious problem.

4. The MMD-CC method is underwhelming at a practical level as well.  First, it requires twice as many samples as Sutherland's MMD.  Despite that limitation, it performs worse in Table 1.

5. For the CIFAR-10-vs-CIFAR-10.1 experiment, the proposed feature generators were trained using ImageNet (see line 160).  This is problematic for several reasons.  First, using an external datset for OOD detection is a form of outlier exposure that violates the main claim of the paper stated on line 45.  Second, the other methods under test do NOT use an external datset, making the comparison unfair.  Third, the use of an external dataset goes against the instructions on line 94, which explicitly state that SimCLRv2 is to be trained on "in-distribution samples." Fourth, alhough the authors wrote that "a fair comparison is difficult," that's not true; they could have easily trained their feature generator on the CIFAR-10 inlier dataset and it would have avoided all these problems.

6. The authors claim a weakness of CSI is that it can't detect adversarial perturbations, but in doing so the authors are conflating two entirely different versions of the OOD problem.  The first version is where one is given a trained supervised classification network and asked to detect samples different from those used to train it.  In this problem, we start with a network and so adversarial perturbations are well defined.  The second version (solved by CSI) is where one is given an inlier distribution and are asked to detect samples different from it.  Here, there is no network, and thus no adversarial perturbations, and so it is unfair/illogical to claim that CSI is "not well suited for adversarial detection." CSI solves an entirely different problem.

7. The section "detection of distribution shifts from a small number of samples" hides the fact that a huge number of samples are required to train the feature generating network.  In fact, by training on an external dataset, the authors are implementing a form of "outlier exposure" that goes against the main requirement stated on line 45.

8. Critical information about the CADet method (the main contribution according to the title/abstract) is missing, making it impossible to fully evaluate the paper.  In particular, the similarity function "s" in (5) is never specified.  On line 195 it says "self-supervised contrastive learning trains a similarity function," but that is incorrect: SSCL trains a feature generating network that maps an input "x" to a set of features "z," as specified on line 99.  SSCL does NOT train a similarity function that maps two inputs "x1" and "x2" to a scalar.

9. The choice to linearly combine the m^in and m^out scores in (6) is heuristic, and probably highly suboptimal.  The framework of multiple hypothesis testing gives principled ways to combine scores.  For more on multiple hypothesis testing, see Candes' STATS 300C lecture notes.

10. The CSI method by Tack et al. is "the closest work in the literature" according to line 86. Yet the authors curiously omit it from all the numerical experiments, choosing much easier competitors.

**Questions:**

1. Regarding the perturbations in Table 2, they are never clearly defined.  Are these perturbations of the Supervised ImageNet-trained classifier used in the bottom two rows?

2. There seems to be a mistake in the contrastive loss equation (1), since the "j" in the numerator is undefined.  Should the numerator be "u_{i,i}"?"

3. There seems to be a mistake in the CADet definition (5).  The denominator of m^out contains the number of terms in the numerator sum, but the denominator of m^in does not.  Should the latter be "ntr(ntr-1)"?

**Limitations:**

yes

---

> ### Author Rebuttal · Authors · 2023-08-09
>
> 1a. We consider that works that tuned their methods on calibration OOD datasets and evaluated on different test benchmarks are still tuning on OOD samples since these test and calibration benchmarks typically involve similar types of distribution shifts, applied from similar inlier distributions. We searched extensively for works that truly do not have any outlier contamination and found very few of them, such as the excellent ViM.
>
> b. While the method in Sohn et al. is interesting, their method relies on learning the distribution of representations, and do not leverage transformations at test time. We will cite their work but we believe it is not more closely related than CSI.
>
> c. For MMD, the p-value we compute are not conformal p-value. They're classic p-values: under the null hypothesis, the scores are identically distributed and thus their ranks are uniformly distributed. Interestingly, in L224-L227, the p-value we compute is indeed a conformal p-value, and we admit its dependence on $X_{val}$ L227. However we only use it as a computation trick to compute the AUROC more efficiently. We verified independently that laborious AUROC computation differ at most in 0.02% in AUROC scores from the ones we report.
>
> d. We were not aware of that specific work and will cite it.
>
> 2. See point 1.c. above. The only part where we compute conformal p-values is in fact just a (valid) computation trick for AUROC measurement. In MMD experiments, the ranks are not computed wrt a separate validation set, and thus there is no such set it could hold marginally on. This is independent from the framework of conformal p-values, and instead relies on the much simpler (and older) framework of p-values.
>
> 3. What does the reviewer mean by "n_perm permutations are not independent". The permutations are chosen at random and are of course iid. This is the same principle as the original MMD method. If n_perm is larger than the number of possible permutations, certain permutations will occur multiple times. There is no problem with that. Each score is computed with a random permutations of variables that, under the null hypothesis, are iid. This is sufficient to guarantee that the rank of the scores are uniformly distributed under the null hypothesis. This is aligned with the classic MMD method.
>
> Compared to "Learning deep kernels for non-parametric two-sample tests" from Sutherland, we even corrected a small mistake in p-value computations (see L131-L139), which made their p-values technically invalid.
>
> 4. As we argue in L144, the benefits of MMD-CC are expected to be significant with small sample sizes. This is confirmed in Table 2 where MMD-CC convincingly outperforms MMD. MMD-CC is an alternative we propose, and we never claim it is always preferable to MMD. It performs better in certain settings (few samples) which is valuable even if it doesnt outperforms prior works in every scenario. Regarding the number of samples required, we only need twice as many samples from one of the two distributions. In practice, we often have access to a much larger number of inliers, so this constraint is irrelevant. When we don't, MMD is likely preferable.
>
> 5. The claim in L45 refers to CADet (not MMD), which is trained on ImageNet and **always** evaluated with ImageNet as inlier distribution. Thus the reviewer's observation does not contradict L45. We agree the previous methods do not use an external dataset, and thus that these results should be considered with caution, and it is specified at L60-L61. The problem is that existing methods do not use inlier training data either. So even if we trained on CIFAR as the reviewer suggested, the comparison would still not be fair. We believe to be the first work distinguishing CIFAR10 and 10.1 with such confidence, with or without external data.
>
> 6. As stated L242-L243 the attacks are computed against a fixed pre-trained network. Therefore, they are black-box attacks and all methods are evaluated against the same images. It is well known that adversarial attacks transfer relatively well across convolutional networks. Since it is a fixed distribution shift, we don't think the reviewer's comment applies.
>
> 7. Again, L45 explicitly refers to CADet, not MMD. We believe for hypothesis testing on CIFAR10.1 vs CIFAR10, outlier exposure or contamination does not make sense. The goal is not to evaluate detection metrics on CIFAR10.1 and claim that such metrics will generalise to other out-distributions, but simply to find that these two specific distributions are indeed different. Anything is fair as long as the p-values are valid.
>
> 8. As specified on L100, the similarity function we refer to is the cosine similarity on top of learned representations. In the training of SSCL, the representations only appear through the intermediary of cosine(z1, z2). Thus it is not incorrect to claim that SSCL learns a similarity function.
>
> 9. It is indeed heuristic. Multiple hypothesis testing is also a valid choice to combine $m_{in}$ and $m_{out}$, but it is often highly suboptimal too. For instance, when introducing random scores with identical distribution in and out of distribution, the performance drop sharply. We could have investigated multiple options to combine scores but then how would we choose between them without using OOD samples ? Thus we opted for a simple heuristic.
>
> 10. Similarly to "OpenOOD: Benchmarking Generalized Out-of-Distribution Detection", we failed to scale CSI to ImageNet sizes.
>
> Q1. The perturbations are defined later in part 5.1 for the CADet experiment which is an issue. We will clarify in section 4.2. Indeed, the attacks are computed against the supervised backbone. Effectively, PGD w/ supervised backbone can be seen as white box, and with the self-supervised backbone as black box.
>
> Q2 and Q3. The reviewer is correct in both case and we will correct these mistakes. Note that for (5) it is not impactful since the scores are rescaled to equalize variance anyway.

---

> > ### Comment · Reviewer_hQ4Y · 2023-08-16
> >
> > Thank you for your rebuttal, which clarified several points in the paper that I missed when reviewing.  As suggestions to improve the final version (if accepted):
> >
> > 1c. The paper would benefit from describing the p-value after L225 as a conformal p-value and citing relevant literature.
> >
> > 3\. The paper would benefit from an explanation/proof of why the proposed MMD-CC test outputs valid p-values.
> >
> > 8\. I still think it is confusing to say that SSCL trains a similarity function when the similarity function (a cosine) is fixed.  Some rewording would help.
> >
> > 10\. Regarding the lack of comparison to CSI, I still believe that its absence significantly weakens your paper.  The OpenOOD paper says only that it took longer than 48 hours to evaluate.  In that work, this excuse is acceptable because they compare dozens of methods on 9 different tasks. But your paper has far fewer methods and fewer tasks, and you claim that CSI is the closest approach in the literature. So I don't think it's acceptable to omit CSI. Furthermore, your paper gives no justification for omitting CSI (only your rebuttal does).
> >
> > In any case, I have modified my review as follows:
> > soundness = 3,
> > presentation = 3,
> > contribution = 3,
> > overall = 5

---

> > > ### Author Response · Authors · 2023-08-18
> > >
> > > We thank again the reviewer for their time reviewing our submission and consideration of our rebuttal.
> > >
> > > We will incorporate the suggested improvements. Regarding MMD-CC, we will add an appendix section with a formal proof of validity of computed p-values. Regarding the similarity function, we do not mean the cosine alone, but the composition of the cosine with the representations, that is, $f(x_1,x_2)=cosine(h(x_1),h(x_2))$ where $h(x)$ computes the representations for image $x$. Thus this function is not fixed, and admits the same parametrisation as $h$. We will add a full sentence specifying precisely our terminology.
> > >
> > > For CSI, we agree with the reviewer that it appears curious to omit evaluation when we describe their method as the closest in the literature. We will add a section in the appendix to give additional details on previous works (that would not fit reasonably in the related work section), such as our specific implementations and hyperparameter tuning for comparisons (as suggested by reviewer X9Xp), but also to justify our choice not to include CSI.
> > >
> > > In term of computations, scaling CSI to ImageNet with ResNet50 is not just longer than 48h. The training loss of CSI requires, on top of the usual contrastive transformation, to also augment each sample with 4 rotations (0, 90, 180, 270 degrees). The number of forward passes and backprops to perform for each step is effectively multiplied by 4. Training a ResNet50 on ImageNet with SimCLR took us over a week of training on 8 A100. Thus multiplying step complexity by more than 4 is not reasonably feasible with our computational means. This is without taking into account the addition of an extra linear layer + softmax objective, and the memory overhead which requires reducing the number of samples per batch (which, according to both original SimCLR papers, lead to performance drops). At inference time, CSI requires to compute nearest neighbor (in term of L2 distance between representations) wrt the whole training set. Of course, this step is very expensive with ImageNet. One can consider only a subset of the training set, but with a 1000 different classes, one can expect a large performance drop if the subset is reasonably sized. Finally, as stated L91-L93, CSI multiply their scores by the norm of representations. Unsurprisingly, performing detection based on representation norms alone leads to worse-than-random performances on successful adversarial attacks, which tend to have higher representation norms than inliers. Thus CSI is not reasonable for adversarial detection (though it would be very interesting to compare to CSI for novel class detection, if our hardware permitted it).

---

### Official Review · Reviewer_1ZdX · 2023-06-26

**Soundness:** 3 good
**Presentation:** 3 good
**Contribution:** 3 good
**Rating:** 5
**Confidence:** 2

**Summary:**

This paper studies the OOD detection of new classes and adversarial attacks with learned similarities via self-supervised contrastive learning in conjunction with MMD two-sample test. To enable MMD applicable for OOD detection on single samples, they improve the idea by using augmentations to create a set. Experimental results on new classes and adversarial examples detections show that their method can perform favorably against previous arts.

**Strengths:**

1. The paper is well-written and well-organized.

2. The paper does a good job summarizing preliminary ideas and providing detailed settings reproducing the overall algorithm.

3. The paper provides helpful discussions regarding the efficiency of the method.

**Weaknesses:**

1. According to Table 3. it seems that the proposed method does not show superior performance against previous arts when using supervised training. Discussions regarding this matter could be included to better justify the method.

2. I'm not an expert on OOD detection. I'm wondering why there are no error bars in the reported results, given that different hyper-parameters could easily improve (or decrease) the performance, and the differences between the compared arts are not that significant reported in Table 3.

3. Some hyper-parameters are empirically selected without proper justifications, especially those regarding augmentation.

**Questions:**

See weakness

**Limitations:**

Limitations have been discussed in the manuscript.

---

> ### Author Rebuttal · Authors · 2023-08-09
>
> We thank the reviewer for their time and feedback.
>
> 1- As mentioned L195-L198, CADet is designed to leverage the similarity function of self-supervised contrastive learning (SSCL). Indeed, the CADet score is computed based on the cosine similarity between the representations of random transforms of a same image (for m_in) or of distinct images (for m_out). Those are the exact quantities that SSCL models are trained with: these models are trained to maximise the similarity between random transforms of a same image, and minimise the similarity between random transforms of two different images. This alignement between the training objectifs of SSCL and the quantities used by CADet is what makes CADet particularly suited for SSCL. Note that on iNaturalist, the supervised backbone still performs better because iNaturalist classes are very close to certain ImageNet classes, and having to differentiate plant species within ImageNet allow supervised classifiers to learn very fine-grained features, which are useful to detect such close OOD examples. On all other benchmarks, SSCL backbones are better suited for CADet.
>
> Since CADet was designed specifically with SSCL in mind, it is predictable that its performance with a supervised backbone, which we report as a mean of comparison, would be inferior. Surprisingly, it is not so far from SotA, ranking only behind Mahalanobis and ViM. Note that while CADet is not SotA on novel classes detection benchmarks, it beats previous works in detecting adversarial attacks (Table 4), and still performs close to best methods on novel classes detection (Table 3) without requiring any change of tuning, and without requiring any training label either. We believe such results are sufficient to warrant publication.
>
> 2- Since it is now generally accepted in the field that detection methods should not assume access to outliers for validation (or they would overfit outlier distributions, without necessarily transferring to other distribution shift types), we refrain from tuning hyperparameters on outliers. Effectively, we use the standard setting of SimCLRv2 for training, and for instance our hyperparameter gamma is autocalibrated on the training set in a deterministic way. As such, it is not clear to us which different hyperparameters values should be used to compute error bars.
>
> Ideally, error bars would be computed wrt k independent trainings (with separate seeds) for the underlying network. However, as self-supervised contrastive learning is very expensive, and we insist on evaluating at a scale that is relevant to the real world, doing such independent runs is outside of our computational means.
>
> 3- The transformations used are the default ones reported in "Big Self-Supervised Models are Strong Semi-Supervised Learners" (SimCLRv2) by Chen et al. Unfortunately, since we refrain from seeing outliers, we do not have any objective ways to tune hyperparameters, and resort to default settings. In the case of SimCLRv2, these hyperparameters were likely optimized to maximise post-finetuning validation accuracy, and thus were not exposed to outliers. In the case of gamma, we use an heuristic (balancing the variance of m_in and m_out) which is likely highly suboptimal, but without access to outliers, we do not see viable alternatives.

---

### Official Review · Reviewer_tQNn · 2023-06-28

**Soundness:** 3 good
**Presentation:** 2 fair
**Contribution:** 3 good
**Rating:** 5
**Confidence:** 4

**Summary:**

The authors propose to handle the OOD detection problem, exploring self-supervised contrastive learning to detect samples from previously unseen classes and adversarially perturbed samples. They use self-supervised contrastive learning with the maximum mean discrepancy (MMD) to test if two sets of samples originate from the same distribution. They propose CADet which takes advantage of the similarity of the contrastive representation/transformation of the same sample. The proposed approach is thoroughly tested in two scenarios: the scenario with previously unseen classes and the scenario with adversarially perturbed samples.

**Strengths:**

+ The problem of OOD detection is very important in a lot of aspects of machine learning, and it's a worthwhile problem to study and research.
+ The proposed approach, CADet, achieves good performance on the benchmarks.
+ The combination of MMD and self-supervised contrastive learning seems to be effective.

**Weaknesses:**

- The work is motivated by the claim that OOD samples will be encountered in the wild when the machine learning models are deployed in the wild, but this work is not tested in the wild, in a smart computing or cyber-physical system setting. This is a minor complaint as it is tested on (solely) datasets, but it is advised that the authors revise the paper to not include such claims.

- The work fails to be evaluated on "fake" OOD samples, or samples that might originate from a different distribution than the training samples but belong in the same classes. For example, the Webcam domain in the Office-31 dataset and the Amazon domain in the same dataset, have the same classes, but distribution-wise, the samples are different. Will CADet consider Webcam's samples of class A to be OOD from Amazon's samples of class A?

- The Related Works section is too high-level: it's evident that the authors are aware of the recent advance in the field of OOD detection, but a more detailed explanation (one sentence or two) for (most of) the mentioned works would be desirable.

**Questions:**

+ What is the intuition behind the fact that self-supervised contrastive learning works well with MMD?

**Limitations:**

+ The paper proposes an effective approach (self-supervised CL + MMD), but there are other mainstream alternatives to MMD such as the KL divergence and Mahalanobis distance. Why do you pick MMD in particular, instead of KL-divergence and Mahalanobis distance (both of which can be adapted to perform on both instance-level and distribution-level)? Is there any comparison on why KL-divergence based and Mahalanobis distance based approaches aren't as effective?

---

> ### Author Rebuttal · Authors · 2023-08-09
>
> We thank the reviewer for their time and feedback.
>
> **Weaknesses:**
>
> 1- The reviewer is correct that we only evaluate on homogenous benchmark. Evaluating in a realistic setting is extremely difficult, as even data collection can pose legal challenges. Nonetheless, the general motivation for trying to detect OOD samples is that they will be encountered in the wild, even though our evaluations are only an approximation of this phenomenon. While this setting is not perfect, it is the standard approach in the hundreds of OOD detection papers published every year.
>
> 2- Even when samples have the same classes, significant covariate shift can lead to misclassification with overconfidence. Besides, the separation into class is arbitrary, and there is no objective reason to consider that covariate shifts are not a change of class (does a picture of a cat and a drawing of a cat belong to the same class ? there is no objective answer, and certain real world systems trained to recognise animals may need to react differently when presented with a drawing). As such, the general formulation of the OOD detection problem is to detect samples that are out-of-distribution, regardless of their semantic classes. While a few works have tried to focus on detecting unknown classes and NOT covariate shifts, in particular "Full-Spectrum Out-of-Distribution Detection" by Yang et al, again, the benchmarks we use are standards in the field.
>
> 3- We had to decrease the size of the related work section to fit page constraints. Another issue is that the literature of OOD detection has been growing very quick, and it is difficult to give a good representation of the MANY recent works on the topic while also giving insights into specific works. We will add a reference to "Generalized Out-of-Distribution Detection: A Survey" by Yang et al. which is an excellent survey of recent works, so that the curious reader can gather more detailed insights into concurrent works.
>
> **Questions:**
>
> 1- That is an excellent question. While traditionally self-supervised contrastive learning is used only to learn useful representations, these representations only intervene in the training loss through the similarity function $s=cosine(z1,z2)$. In particular, self-supervised contrastive learning trains the similarity function $s$ to be close to $1$ on pairs of samples that are semantically close (which is approximated by random transformations of a same image) and close to $0$ on pairs of samples that are semantically far (which is approximated by random transformations of different images). Moreover, since the model does not access training label, it does not discard information that is not directly useful for classification, and does not overfit training classes. In MMD, a kernel is used to estimate the proximity between all pairs of samples within a set. For large dimensional, unstructured data like images, traditional kernels often fall short. However, self supervised contrastive learning was precisely train to identify such proximity. Thus, it is a very natural choice.
>
> **Limitations:**
>
> Mahalanobis as it is used in the OOD detection literatures, works by learning a class-conditional gaussian model of representations. In the case of e.g. CIFAR10 vs CIFAR10.1, on what data would this gaussian model be fit ? Since the neural network was trained on ImageNet, and does not make class predictions, it could only find a large OOD score for both CIFAR10 and CIFAR10.1, making it an inadequate approach. Training on a subset of CIFAR10/CIFAR10.1 test set would also prove difficult given their small sizes. As for learning a gaussian model directly on the raw images, it is well known that it is very unadapted to such high dimensional unstructured data. For KL divergence, a similar approach is generally used: assuming and learning a gaussian distribution to estimate KL divergence between OOD samples and in-distribution. However, this method needs to be performed on the representations of a neural network or other model (as the raw data is too high dimensional and unstructured). The same issues then arise: training on a subset of test datasets is not sample efficient and not realistic in this case, and training on an external dataset (such as ImageNet) will yield distribution estimation that are extremely far from both sample sets.
>
> This is where self-supervised contrastive learning provide a convincing solution: the similarity function learned on imagenet is capable of comparing two out-of-distribution samples z1 and z2 to one another directly, whereas KL or Mahalanobis could only compare z1 and z2 to its training distribution, separately.

---

### Official Review · Reviewer_X9Xp · 2023-07-02

**Soundness:** 3 good
**Presentation:** 3 good
**Contribution:** 3 good
**Rating:** 5
**Confidence:** 4

**Summary:**

The paper proposes Maximum Mean Discrepancy with Clean Calibration (MMD-CC), as an improvement of MMD when the number of samples is small. Moreover, it introduces CADet, a novel anomaly detector for both OOD and adversarial detection inspired by MMD.

**Strengths:**

In line with recent works, unsupervised anomaly detection in deep learning is a relevant research objective. The paper is well-written and proposes a novel approach for anomaly detection, improving existing works on Maximum Mean Discrepancy. Moreover, I appreciated the idea of improving MMD to overcome its limitation of requiring sets of test samples.

**Weaknesses:**

- Experimental setting. I did not understand why for benchmarking adversarial detection you kept ODIN [1] and discarded the Mahalanobis detector. To the best of my knowledge, ODIN is mainly designed for OOD detection, furthermore in [2] the Mahalanobis detector achieves better performance than both LID [3] and ODIN for adversarial detection. Moreover, no adaptive attack was tested [4].
- Strength of the results. In Table 3, CADet has a high variance of the results depending on the training strategy  (iNaturalist vs ImageNet-O) and is not the overall best performing. Moreover, only the AUROC was reported, while other commonly employed metrics are missing (FPR, AUPR). Regarding MMD-CC, it should be better clarified its significance given that it is significantly better only on Table 2, with PGD and small n_samples.
- Missing related works. Recently, other unsupervised anomaly detection (for both OOD and adversarial) algorithms have been proposed, such as [5]. A comparison with such method should be performed.
- Reproducibility. No code was released. Some hyperparameters are missing (e.g., number of neighbours for LID [3], perturbation size for Mahalanobis [2] and ODIN [1]). Given how challenging it is to evaluate and compare OOD detectors, this is an important weakness.
- Architecture scale. Given the existence of other unsupervised methods for anomaly detection ([5]) and the lack of convincing performance improvements over related methods, the computational limitations of CADet are a significant weakness.

[1] Liang, S., Li, Y., & Srikant, R. (2018). Enhancing the reliability of out-of-distribution image detection in neural networks. ICLR 2018.

[2] Lee, K., Lee, K., Lee, H., & Shin, J. (2018). A simple unified framework for detecting out-of-distribution samples and adversarial attacks. NeurIPS 2018.

[3] Ma, X., Li, B., Wang, Y., Erfani, S. M., Wijewickrema, S. N. R., Schoenebeck, G., Song, D., Houle, M. E., & Bailey, J. (2018). Characterizing adversarial subspaces using local intrinsic dimensionality. ICLR 2018.

[4] Tramer, Florian, et al. "On adaptive attacks to adversarial example defenses." Advances in neural information processing systems 33 (2020): 1633-1645.

[5] Raghuram, J., Chandrasekaran, V., Jha, S., & Banerjee, S. (2021). A General Framework For Detecting Anomalous Inputs to DNN Classifiers. Proceedings of the 38th International Conference on Machine Learning, 8764–8775.


**Questions:**

### Suggestions:
Given the amount of OOD and adversarial detectors now available, I would expect a strong validation of a method to prove its effectiveness. I think that the most important concern that should be addressed is how CADet, given its resource limitations, can be a significant improvement over existing unsupervised OOD/adversarial detectors such as [1].

### Typos:
- Line 140, "Additionnally".

[1] Raghuram, J., Chandrasekaran, V., Jha, S., & Banerjee, S. (2021). A General Framework For Detecting Anomalous Inputs to DNN Classifiers. Proceedings of the 38th International Conference on Machine Learning, 8764–8775.


**Limitations:**

The authors addressed the main limitations of their method.

---

> ### Author Rebuttal · Authors · 2023-08-09
>
> We thank the reviewer for their time and feedback.
>
> 1- In [2], the Mahalanobis detector uses a score based on a logistic regression of layer-wise Mahalanobis distances. The regression weights are trained on validation outliers. Training directly a classifier on outliers has long been discarded by the community and makes their comparison unfair and explain their superior performance. This is the reason why we kept ODIN, which is indeed designed as an OOD detector but has been shown in prior works to be somewhat reasonable in detecting adversarial samples, and discarded the detector of [2] which operates in a much easier and less relevant setting where training on outliers is allowed (ODIN does some hyperparameter tuning on outliers but it is a lighter violation as they merely do a grid search on two hyperparameters and show their tuning to transfer well). With that said, for OOD detection, we followed the steps of prior works such as ViM and adapted the Mahalanobis score by only considering it for the last feature layer. Doing this removes the need to train on outliers. We are not aware of any prior works having followed the same process for adversarial detection, but it seems like a natural and logical thing to do. We will immediately start this evaluation, and hopefully will be able to provide results before the discussion phase ends.
>
> Regarding adaptive attacks, it is difficult enough to fit our submission within the page limit. Evaluating against adaptive attacks is a comprehensive subject and would require a dedicated paper. Moreover, while robustness to adaptive attacks is desirable, it is well known that ood detection method are weak against them, and being able to detect attacks in a black box setting is already useful. We believe our method, like its peers, would be weak against adaptive attacks.
>
> 2. We admit CADet is not best performing for novel class detection. However, its performance on adversarial detection alone is impressive and in our opinion could warrant publication. The fact that it is also concurrently close to SotA on novel class detection, without requiring any change of tuning, is a significant plus. Would the reviewer find our results more convincing if we only reported adversarial detection ?
>
> It is true that we only report AUROC. The reason for this is that in our experience, AUROC FPR95 and AUPR typically follow the same pattern, and yields the same rankings across methods. We are not aware of any case where the performance (relative to other methods) vary significantly across metrics. As thus, in our opinion this set of metrics has become redundant and makes results less readable.
>
> MMD-CC is an alternative we propose to classic MMD, which can be advantageous at smaller sample sizes. In Table 2, MMD-CC outperforms MMD on all benchmarks and for all sample sizes (though they are all small), with some ties on iNaturalist.
>
> 3. We will cite and discuss [5]. Note that [5] introduces a meta-algorithm which it experiments with existing methods on small datasets, but scaling e.g. aK-LPE to ImageNet requires significant computation ressources. Since most existing methods already follow the abstraction of their meta algorithm, experimenting against these methods, in a sense, is an experimentation against a specific realization of their meta algorithm.
>
> 4. For LID, we used k=20 as was used by authors on CIFAR and MNIST, and will specify it. We agree that the absence of code is an important issue. We intend to release our code publicly, the only obstacle being that it is pending legal review. While this legal review has been ongoing for an absurdly long time, we have been promised we should get clearance soon, much before camera ready deadlines. We intend to release our code as a fork of OpenOOD, which should make it especially easy to use and reproduce by the community.
>
> 5. As stated above, [5] proposes a meta-algorithm rather than a specific method. They also introduce two specific realisations of their meta-algorithm using scores from previous works. aK-LPE does not scale well either to large datasets, and while we are not absolutely certain, we suspect that the other method of using p-value normalization at each layer *and layer pairs* likely does not scale well to modern architecture either. While CADet has a significant overhead, it is entirely parallelizable (all the overhead is induced by having to do multiple forward passes) and the overhead factor is independent of the architecture. Given its jointly good performances on adversarial detection and novel class detection without ever requiring any labels, we believe it is still an interesting contender that is relevant to the community.
>
> Suggestion: While [1] report clock time including training in Table 9, they fail to report inference time, which is most relevant and is likely to be high. Our reported overhead is purely for inference time.

---

> > ### Comment · Reviewer_X9Xp · 2023-08-11
> >
> > After reading the rebuttal, I have some additional concerns/questions:
> > 1. Your design choices on the Mahalanobis and LID detectors should be better clarified. You should explain that you kept only the last layer for the Mahalanobis distance (following the ViM method); indeed, as you correctly stated in the rebuttal, it makes sense to extend it in the adversarial scenario. Given this new experimental details, I wonder if also for LID you kept only the last layer, since otherwise it would have required to train a classifier. At least to me, these details were non-obvious during the review and should be clarified (at least in the Appendix).
> > 2. The lack of adaptive (or an analysis with various, smaller, perturbations sizes) makes the evaluation on adversarial attacks limited in comparison to other detectors (like JTLA), given also the fact that it is one of the main selling points of CADet. I think this should be (briefly) stated in the conclusions/limitations.
> >
> > Regarding the other issues:
> >
> > 3. I think that the reproducibility concerns, excluding 1 above, are addressed. In particular, I thank the reviewers for detailing their status with regard to the code publication issue.
> > 4. Since no clear difference among the metrics (AUROC, ...) was detected, I agree on favouring readability over completeness.
> > 5. Regarding the JTLA method, I acknowledge that it would have required some adaptations (fitting aklpe on a subset of ImageNet), since it was tested only on smaller training sets. Regarding related works, I would also reference [1], that does not require labelled adversarial and scales to Imagenet. Of course I do not expect further testing on it given the restricted timeframe.
> > 6. Regarding the performance of CADet on novel class detection, I would leave the results section and the overall presentation as it is (CADet as a -generalized- anomaly detector), given my concerns on the evaluation of adversarial attacks.
> > 7. I would state better that MMD-CC shows significant improvements (at a computational cost) mostly on PGD, since in the other cases performs essentially on par with MMD (apart from iNaturalist, n=3).
> >
> > Minors:
> > - citations in methods of table 4.
> > - missing citations of adversarial detectors employed in the results in the related works section, such as LID ([40]).
> >
> > To summarise, after re-reading the work and the rebuttals, I think that the work contains valuable contributions. However, I have still have some doubts on the performance improvements over SOTA (in particular with regard to adversarial attacks), together with the computational limitations. I am willing to increase the rating after the response by the authors to my additional observations.
> >
> > [1] "The Odds are Odd: A Statistical Test for Detecting Adversarial Examples", Roth K. et al., ICML 2019.

---

> > > ### Author Response · Authors · 2023-08-14
> > >
> > > We thank the reviewer for their careful consideration of our rebuttal, and more generally of our submission.
> > >
> > > 1. Indeed, we only compute nearest neighbours based on the last layer, and then directly use it as a detection score. The last layer was found in the original paper ([1], Figure 2) to be the most useful to directly perform detection. However, as the reviewer noted, this is not specified anywhere in our draft and we simply refer to the original paper which mainly investigates the version of their algorithm training a classifier on all layer scores. Similarly, while it has been fairly common in the field to use the Mahalanobis distance of the last layer as a detection metric, we cite the original paper without specifying how we perform computation. We will add in appendix a section focusing on a precise description of our implementation of recent methods, with full hyperparameters and such adaptations.
> > >
> > > 2. We will add a subsection in the limitation section focusing on discussing adaptive attacks. Our viewpoint is that defence methods, including detection-based, are always weak against adversarial attacks. [2] showed that many recent decent methods claiming robustness to adaptive attacks are actually weak against well-designed adaptive attacks. In general, recent work can only show the robustness of proposed methods against adaptive attacks of their design. It has become a consistent trend in the literature to subsequently observe that they are in fact not robust to properly design adaptive attacks. Given these elements, we find that even if we found our method to be robust against an adaptive attack of our design, it would be insufficient to claim robustness to such type of attacks. Realistically, it is safe to assume that CADet is not robust to adaptive attacks, like most recent works, and it is nearly impossible to prove that other detection methods possess such robustness anyway. Nonetheless, there is still practical motivation to seeking such detectors: while it is well known that adversarial attacks transfer relatively well across CNN architectures, adaptive attacks against detectors require very specific knowledge of the attacked detector. As such, attacking an unknown CNN-based classifier remains fairly easy, while trumping a detector requires knowledge of the model, adding protection.
> > >
> > > 5- We were not aware of this work, which indeed seems extremely relevant. We will investigate it in detail, cite it, and if possible produce comparisons before camera-ready deadline -- should our submission be accepted.
> > >
> > > Minors: we will fix these issues
> > >
> > > [1] Characterizing Adversarial Subspaces Using Local Intrinsic Dimensionality, *X Ma et al.*
> > >
> > > [2] On Adaptive Attacks to Adversarial Example Defenses, *F Tramer et al.*

---

> > > > ### Comment · Reviewer_X9Xp · 2023-08-15
> > > >
> > > > 1. Thank you for the additional details, that I think are necessary to have a full understanding of your experimental setting. Indeed, in my experience, I always considered the multi-layer implementation of both LID and Mahalanobis.
> > > > 2. Given that your framework is not meant to be a defence against adversarial attacks, but rather a new perspective on unsupervised (generalized) OOD detection, I acknowledge that adaptive attacks can be skipped. Although, given the literature in this regard and the expected experimental settings, I would briefly discuss it as you plan to do.
> > > >
> > > > The final version of this paper will require an important revision with regard to reproducibility, comparison with SOTA, and clarity. But the authors addressed all of my concerns and demonstrated their willingness to improve their work. Consequently, I will raise my soundness and contribution scores from 2 to 3 and my rating from 3 to 5.

---

> > > > > ### Author Response · Authors · 2023-08-18
> > > > >
> > > > > We would like to thank again the reviewer for their time reviewing our submission and considering our rebuttal. We will make sure to carefully incorporate all suggested improvements, and promise the code will be fully released by publication date, should our submission be accepted.

---

### Official Review · Reviewer_kyFG · 2023-07-27

**Soundness:** 3 good
**Presentation:** 4 excellent
**Contribution:** 3 good
**Rating:** 6
**Confidence:** 3

**Summary:**

The paper presents a novel method for Out-of-Distribution (OOD) detection called CADet (Contrastive Anomaly Detection). The authors leverage self-supervised contrastive learning and the maximum mean discrepancy two-sample test(MMD) to assess whether two sets of samples have been drawn from the same distribution. The method is designed to detect OOD samples from single inputs and performs well on both label-based and adversarial detection benchmarks, without requiring access to any OOD samples nor previous classes to train or tune the method.

**Strengths:**

1. The authors propose a novel method for OOD detection that does not require access to neither in distribution samples nor OOD samples to train or tune the method.
2. They use similarity functions learned by self-supervised contrastive learning with MMD to assess the distribution even with few shot images.
3. Paper shows the method outperforms current methods in adversarial detection tasks while performing well on label-based OOD detection.


**Weaknesses:**

I do not see major weaknesses for this paper as of now. I will be updating this section if need be later.

**Questions:**

1. I understand ResNet50 is a popular architectures that has been used for OOD and adversarial benchmarks. But there has been significant developments in those areas too, did you have a chance to see how would the method perform on other architectures beyond ResNet50?

**Limitations:**

Authors have done a good job presenting the limitations of their paper.

---

> ### Author Rebuttal · Authors · 2023-08-09
>
> We thank the reviewer for their time and feedback.
>
> **Q1:** It is a very pertinent question and recent works such as "ViM: Out-Of-Distribution with Virtual-logit Matching" by Wang et al. shows more recent architectures like ViT have distinct and promising features when it comes to OOD detection. Unfortunately, self-supervised contrastive pre-training is expensive and our computation resources did not allow us to experiment with larger architectures, as admitted in L292-L301.
>
> We seem to have noticed the reviewer have reduced their score from the initial review and would like to inquire if they have any novel question or concerns we might address ? If such decision was motivated by other reviews, we invite the reviewer to also consider our rebuttals to these reviews.

---

### Decision · Program_Chairs · 2023-09-21

**Decision:**

Accept (poster)

**Comment:**

The paper introduces an out-of-distribution detection algorithm based on MMD and contrastive learning. Reviewers appreciated the novelty of the method but had initial concerns on the experimental evaluation, strength of results and discussion of related work. The authors' responses and following discussion addressed most of these concerns to the satisfaction of the reviewers. Following the discussion, reviewers unanimously lean towards acceptance. The authors should ensure all promised revisions to the paper are included in the camera ready version.